# QiMeng-SALV: Signal-Aware Learning for Verilog Code Generation

Yang Zhang[1,2]    Rui Zhang[1]    Jiaming Guo[1]    Lei Huang[1,2]    Di Huang[1]
Yunpu Zhao[1,3]    Shuyao Cheng[1]    Pengwei Jin[1,2]    Chongxiao Li[1,2]
Zidong Du[1]    Xing Hu[1]    Qi Guo[1]    Yunji Chen[1,2*]

[1]State Key Lab of Processors, Institute of Computing Technology, CAS
[2]University of Chinese Academy of Sciences
[3]University of Science and Technology of China

https://qimeng-iprc.github.io/QiMeng-SALV/

## Abstract

The remarkable progress of Large Language Models (LLMs) presents promising opportunities for Verilog code generation which is significantly important for automated circuit design. The lacking of meaningful functional rewards hinders the preference optimization based on Reinforcement Learning (RL) for producing functionally correct Verilog code. In this paper, we propose Signal-Aware Learning for Verilog code generation (QiMeng-SALV) by leveraging code segments of functionally correct output signal to optimize RL training. Considering Verilog code specifies the structural interconnection of hardware gates and wires so that different output signals are independent, the key insight of QiMeng-SALV is to extract verified signal-aware implementations in partially incorrect modules, so as to enhance the extraction of meaningful functional rewards. Roughly, we verify the functional correctness of signals in generated module by comparing with that of reference module in the training data. Then abstract syntax tree (AST) is employed to identify signal-aware code segments which can provide meaningful functional rewards from erroneous modules. Finally, we introduce signal-aware DPO which is optimized on the correct signal-level code segments, thereby preventing noise and interference from incorrect signals. The proposed QiMeng-SALV underscores the paradigm shift from conventional module-level to fine-grained signal-level optimization in Verilog code generation, addressing the issue of insufficient functional rewards. Experiments demonstrate that our method achieves state-of-the-art performance on VerilogEval and RTLLM, with a 7B parameter model matching the performance of the DeepSeek v3 671B model and significantly outperforming the leading open-source model CodeV trained on the same dataset. Our code is available at https://github.com/QiMeng-IPRC/QiMeng-SALV.

## 1 Introduction

Circuit design is inherently complex and time-consuming, particularly with respect to Hardware Description Language (HDL) such as Verilog. Automating the HDL coding process is of paramount importance as it significantly enhances the efficiency of circuit design. The remarkable progress of

---

*Corresponding author. Contact: {zhangyang22s2,cyj}@ict.ac.cn

39th Conference on Neural Information Processing Systems (NeurIPS 2025).

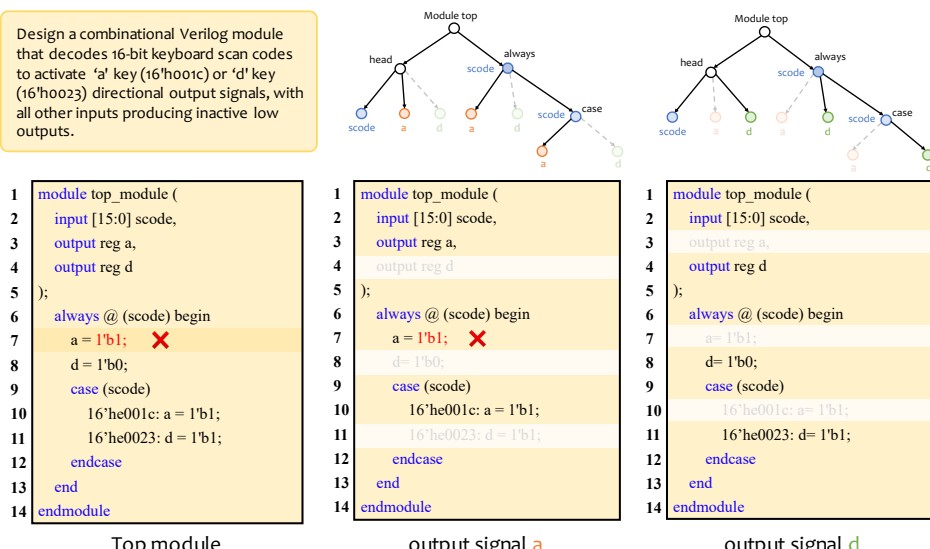

Figure 1: Due to Verilog's characteristics of relatively independent and parallel signals, we can extract the code implementation of a specific signal through AST. Here there are two output signals, we can separately extract their related code implementations. Among them, the implementation of signal a is incorrect while signal d is correctly implemented. We can utilize the code implementation of the correct signal d to provide functional correctness rewards for RL.

Large Language Models (LLMs) in program code generation, such as Python, presents promising opportunities for Verilog code generation. To equip LLMs with superior code generation capabilities, post-training procedures are essential, primarily comprising Supervised Fine-Tuning (SFT) and preference-based Reinforcement Learning (RL). SFT provides foundational programming language syntax comprehension and domain-specific knowledge, while RL optimization techniques, like Proximal Policy Optimization (PPO) [1], Group Relative Policy Optimization (GRPO) [2], and Direct Preference Optimization (DPO) [3], enhance the model's ability to produce functionally correct code.

Currently, most Verilog code generation methods primarily focus on the SFT stage, particularly through dataset preparation [4, 5, 6]. Few methods [7, 8] employ RL through using code structure similarity with reference code as the reward. However, relying solely on code structure similarity as a reward has inherent limitations, since a module may has many different Verilog implementations which exhibit structural differences but are functionally correct. Thus, it is more reasonable to use functional correctness as the reward in RL, which requires the model to sample functionally correct module implementations to provide the reward. However, the lack of high-quality SFT training data frequently results in inadequate initial code generation capability, obstructing the acquisition of meaningful functional rewards and consequently impairing RL optimization performance. Thus, enhancing the availability of actionable functional rewards to facilitate effective RL optimization remains an open research challenge.

In contrast to high-level program language that defines sequential execution behavior, Verilog code specifies the structural interconnection of hardware gates and wires, granting inherent independence between different output signals. This characteristic enables scenarios where individual signal implementations remain functionally correct despite errors in the overall module implementation. As shown in Fig.1, the faulty implementation of the signal a results in total malfunction of the top module, whereas the signal d maintains proper functionality. Due to Verilog's inherent signal independence and parallel processing nature, we can easily extract a certain signal implementation from the top module through abstract syntax tree (AST). Such partially valid signal-level implementations can be leveraged within RL frameworks to derive meaningful functional rewards, thereby expanding the effective sample space for RL optimization.

Based on the above analysis, in this work, we propose Signal-Aware Learning for Verilog code generation (QiMeng-SALV) by leveraging code segments of functionally correct output signal to optimize RL training. The key insight of QiMeng-SALV is to extract verified signal-aware

implementations in partially incorrect modules, so as to enhance the extraction of meaningful functional rewards. Specifically, to verify the functional correctness of signals, some random test inputs are generated to compare signal-level outputs between generated module and reference module in the training data. Additionally, abstract syntax tree (AST) analysis is employed to precisely identify preferred and dispreferred code segments, thus these partially correct signal-level implementation can be utilized to generate meaningful functional rewards. Finally, to enable the model to learn correct signal implementations from erroneous modules, we introduce signal-aware DPO which is based on DPO, a method renowned for its computational efficiency, rapid convergence, and strong performance. Unlike standard DPO that computes probabilities for all module code tokens, signal-aware DPO exclusively calculates token probabilities for the correct signal-level code segments from the preferred and dispreferred samples, thereby preventing noise and interference from incorrect signals during RL training. Crucially, the proposed QiMeng-SALV expands the effective training dataset by incorporating all modules containing any valid signal implementations, regardless of overall module correctness. This represents a fundamental shift from conventional module-level to fine-grained signal-level optimization in RL-based Verilog code generation, addressing the issue of insufficient functional rewards in RL training. To the best of our knowledge, this is the first fine-grained signal-level reinforcement learning algorithm designed for Verilog code generation.

Comprehensive evaluations demonstrate that QiMeng-SALV establishes new state-of-the-art results on both VerilogEval [9, 10] and RTLLM benchmarks [11, 12]. The 7B-parameter model achieves remarkable 62.6% pass@1 and 75.1% pass@5 accuracy on RTLLM v1.1, not only matching the performance of the 671B-parameter DeepSeek-V3 [13], but also delivering substantial improvements over CodeV [4], the leading open-source model of comparable scale. Ablation studies further validate the substantial impact of filtering incorrect signal segments, confirming the efficacy of QiMeng-SALV's design.

## 2 Related Work

### 2.1 RTL code generation

Recent years have witnessed remarkable achievements of Large Language Models (LLMs) in software code generation tasks, prompting both academia and industry to actively investigate their potential applications in Hardware Description Language (HDL) code generation [14]. This novel research frontier has achieved significant progress, unveiling promising paradigms for hardware design automation.

Instruction tuning has emerged as a pivotal methodology for enhancing LLMs' task adaptability, owing to its methodological simplicity and demonstrated performance gains [4]. This technique shows broad application prospects for automated HDL generation. However, compared to software programming languages (such as Python, C++), the domain-specific expertise of hardware design languages and the scarcity of high-quality training data result in current LLMs' performance in HDL generation tasks being significantly inferior to their performance in software programming languages [15]. Therefore, many works focus on the construction of hardware language datasets [6, 5, 12, 4] and employ Supervised Fine-Tuning (SFT) strategies to enhance LLMs' HDL generation capability. However, the heterogeneous quality of training data, whether scraped from public sources or synthesized by advanced LLMs, remains a critical bottleneck, compromising both validity and reliability. This has spurred innovative approaches incorporating verification frameworks and feedback mechanisms during model training/inference to overcome performance bottlenecks and improve RTL generation quality [16, 14, 17].

### 2.2 RL Training for LLM

The effectiveness of supervised fine-tuning (SFT) in hardware description languages is fundamentally constrained by data quality and diversity, particularly in specialized domains where conventional approaches often fail to meet professional requirements[18]. This limitation has driven interest in reinforcement learning (RL) techniques that can enhance model capabilities beyond what SFT alone can achieve.

Reinforcement Learning from Human Feedback (RLHF)[19] is among the earliest and most impactful approaches for aligning language models with human intent with Proximal Policy Optimization[1].

Subsequent work [20, 21, 3, 22, 23] has focused on developing more efficient alternatives, with Direct Preference Optimization (DPO)[3] emerging as particularly impactful due to its simplicity and stable training process. The recent Group Relative Policy Optimization (GRPO) approach[2] further advanced this direction by eliminating the need for explicit reward modeling, demonstrating particular success in RL-based post-training optimization for higher-performance LLMs.

In the domain of Verilog generation, current RL-based training approaches face two key challenges: (1) highly rely on structural similarity metrics [7, 8] that cannot properly evaluate functional equivalence between different implementations, and (2) the scarcity of high-quality SFT data limits the initial model capability needed for effective RL optimization. While recent work has explored various reward formulations [24, 25, 7] and feedback mechanisms [26, 27, 17], these approaches often introduce evaluation biases, limiting their reliability in assessing functional correctness. VeriPrefer [28] mitigates such biases by employing module-level functional rewards during reinforcement learning. Our approach differs fundamentally: we adopt signal-level functional rewards, allowing partially correct modules to still provide valuable feedback during training when they contain correctly implemented signals. Consequently, our method delivers denser and more informative reward signals, leading to superior reinforcement learning performance.

# 3 Method

## 3.1 Framework Overview

In this work, we propose an innovative training methodology named QiMeng-SALV, which is designed to leverage correct signal-aware implementations within erroneous module codes for Verilog code generation. We start from a naive code generator, which is typically fine-tuned using a Verilog training dataset $D = \{(x, y)\}$ based on a general-purpose LLM, where $(x, y)$ is the description-code pair. The naive code generator samples $k$ candidate module code implementations $E = \{y_1, y_2, ..., y_k\}$ given each module design prompt $x$. Our primary objective is to extract and learn correct signal-aware implementations from these candidates for improving RL optimization. Here we employ DPO as our RL optimization approach, a technique widely recognized for its computational efficiency, fast convergence, and robust performance.

Our training methodology comprises three stages: 1) Signal-aware Verification: By generating random input signals and comparing the output signals between the generated module and the reference module in the training set, we identify correct output signals to obtain preference dataset $P = \{(y_w, y_l, c)\}$, where $y_w$ and $y_l$ represent preferred and dispreferred module code respectively, $c$ represent the contrast signal which is correct in $y_w$ and incorrect in $y_l$. 2) Signal-aware Code Extraction: Leveraging abstract syntax tree (AST) analysis, we establish signal dependency graphs and isolate relevant preferred and dispreferred code segments $(S_w^c, S_l^c)$ corresponding to the contrast signals $c$ from $y_w$ and $y_l$, respectively. 3) Signal-aware DPO training: By computing token probabilities only for the preferred and dispreferred code segments related to contrast signals, we employ DPO to enable the model to learn correct signal implementations, even when the entire module implementation is erroneous. The following sections provide detailed explanations of each stage.

## 3.2 Signal-aware Verification

Enabling RL to train models that generate functionally correct code fundamentally requires functional reward derived from verified implementations. This makes functional correctness verification a critical challenge when applying RL to LLM-based code generation, especially for hardware description languages such as Verilog. Conventional verification methods depend on testbenches to assess module functional correctness, but the lack of such testbenches in training datasets substantially restricts the capacity to provide meaningful correctness feedback during RL training.

Our approach addresses this limitation through an automated signal verification process for generating functional reward. Considering the functional reward is provided by a preference dataset in DPO, we therefore establish the preference dataset based on automated functional verification by three steps: 1) generating comprehensive random input stimuli, 2) verifying generated signals by comparison with reference modules, 3) collecting preference dataset from the verifying results of generated signals, as depicted in Fig.2 (b).

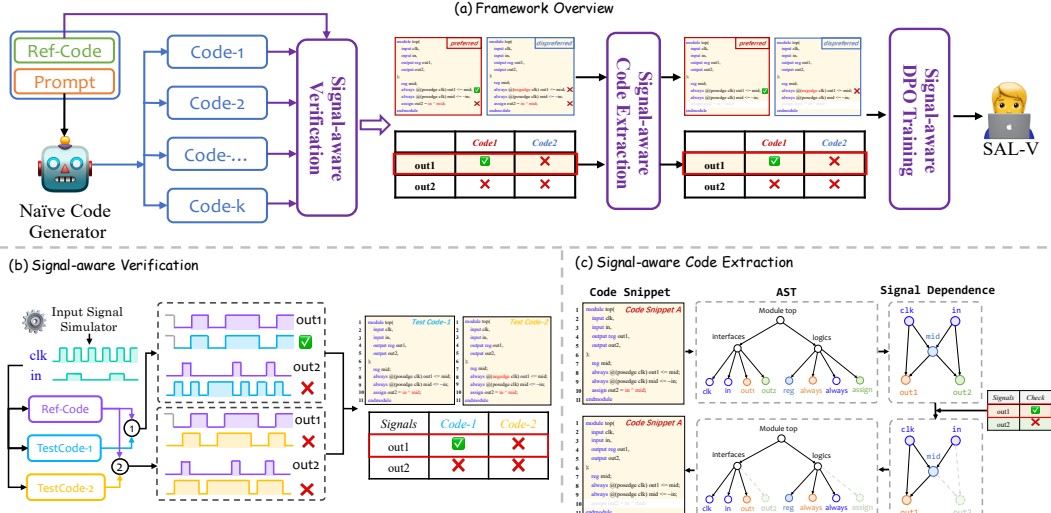

Figure 2: Overview of the QiMeng-SALV framework. a) The proposed QiMeng-SALV comprises three stage: signal-aware verification, signal-aware code extraction, and signal-aware DPO training. b) In signal-aware verification stage, verification is performed by analyzing output signal discrepancies between generated modules and their reference counterparts, allowing precise identification of correctly functioning output signals. c) In signal-aware code extraction stage, AST parsing reveals critical dependencies between output signals and intermediate signals to obtain relevant preferred and dispreferred code segments pertinent to the contrastive signals.

First, we use Yosys [29] to analyze the reference module head, automatically generating $N$ sets of input signals at equal time intervals. Second, the input signals are simultaneously applied to the reference module $y$ and generated modules $E = \{y_1, y_2, ..., y_k\}$. By comparing the output signals produced by the generated modules with the corresponding output signals from the reference module, we can verify the correctness of generated modules, and obtain the pairs $\{(y_1, c_1), (y_2, c_2), .., (y_k, c_k)\}$, where $c_i$ is the correct output signals for generated module $y_i$. Notably, only output signals exhibiting complete matching across all $N$ sets of input signals are verified as correct, while any discrepancies result in classification as erroneous signals. Third, to select preference samples suitable for DPO training, we choose sample pairs that there exists a set of contrast signals that is correct in the preferred sample but incorrect in the dispreferred sample. This can be formally expressed as:

$$P = \{(y_w, y_l, c)\}, \exists c \in c_w \wedge c \notin c_l, \tag{1}$$

where $P$ denotes the preference dataset, $c$ represents the contrast signals, $c_w$ and $c_l$ represent the correct signals in the preferred sample $y_w$ and dispreferred sample $y_l$ respectively.

## 3.3 Signal-aware Code Extraction

In RL-based code generation, there exist two typical feedback mechanisms: outcome feedback and process feedback. Compared to outcome feedback which only indicates whether the final output is correct or not, process feedback can precisely identify which parts of the code implementation are correct, thereby providing a more dense reward for more effective learning. Therefore, by extracting the specific code implementation corresponding to the output signal and adopt the RL method based on process feedback is of great significance for code generation.

In contrast to program code, which sequentially describes execution logic, Verilog code characterizes the interconnection between wires and gates. This indicates that signal-describing code blocks are largely self-contained, allowing individual signal implementations to maintain functional accuracy even when the broader module contains errors. These partially correct signal-level implementations can be utilized in RL frameworks to generate meaningful functional rewards, thus increasing the viable sample space for RL optimization.

Based on the above analysis, in the signal-aware code extraction stage, for given target signals, we parse the Abstract Syntax Tree (AST) to establish dependency relationships between signals, and

then extract the complete code segment corresponding to the target signals, as depicted in Fig.2 (c). Specifically, given the module code of the target signal, we use Yosys [29] to parse the module code and obtain the AST. Subsequently, we analyze the AST to derive the dependency relationships between all the output signals and intermediate signals defined in the module code, forming a signal topology graph. By analyzing this topology graph and performing backward traversal from the target output signal leaf nodes, we obtain the dependent signals of the target signal. Based on the locations of these dependent signals in the AST, we retain their corresponding code segments, thereby obtaining the complete code segment of the target signal. To facilitate the model's learning by using code segment of contrast signal during DPO training, given the contrast signals $c$, we isolates the relevant code segment $S_w^c$ from the preferred sample $y_w$. Meanwhile, to enable comparison with the preferred sample, we also extract the code segment $S_l^c$ of contrast signal $c$ from the dispreferred sample $y_l$.

### 3.4 Signal-aware DPO Training

In this stage, we employ the preferred and dispreferred code segments related to contrast signals in DPO training to strengthen the model's ability of generating functional correct signal implementations. Standard DPO approaches uniformly increase the likelihood of all tokens in preferred code samples while suppressing all tokens in dispreferred samples. This paradigm inherently assumes the absolute correctness of the preferred samples, which is often violated in practice due to suboptimal SFT models. Because of the limited training data of Verilog code implementations during the SFT stage, the SFT models frequently fail to generate completely accurate code samples. Our key insight recognizes that while generating completely accurate modules remains challenging, we can still leverage partially code segments of functionally correct output signal from incorrect modules to optimize DPO training.

Therefore, we introduce signal-aware DPO, an improved DPO algorithm for Verilog code generation, which increases the probability of correct signal-related code in preferred samples while decreasing the probability of erroneous signals in dispreferred samples. To achieve this, during loss computation, we only calculate the token probabilities for the contrasting signal-related code segments in both preferred and dispreferred samples, while ignoring the probabilities of other code segment tokens. Formally expressed as:

$$\mathcal{L}(\pi_\theta; \pi_{\text{ref}}) = -\mathbb{E}_{(x,y_w,y_l,S_w^c,S_l^c)\sim\mathcal{D}} \left[ \log\sigma\left(\beta\sum_{y_t\in S_w^c}\log\frac{\pi_\theta(y_t|y_{w,<t},x)}{\pi_{\text{ref}}(y_t|y_{w,<t},x)} - \beta\sum_{y_t\in S_l^c}\log\frac{\pi_\theta(y_t|y_{l,<t},x)}{\pi_{\text{ref}}(y_t|y_{l,<t},x)}\right)\right], \quad (2)$$

where $y_w$ and $y_l$ represents the preferred and dispreferred sample respectively. $S_w^c$ denotes code segments related to contrast signals $c$ in preferred samples, while $S_l^c$ represents the same signal related segments in dispreferred samples. $\pi_\theta$ is the policy model to be optimized and $\pi_{\text{ref}}$ is the reference model used for regularizing $\pi_\theta$ with Kullback-Leibler divergence and $\beta$ is a constant to control the degree of regularization. Through the objective function in Eq. (2), signal-aware DPO shift DPO's learning focus from entire modules to individual signals, refining the learning granularity and increasing the quantity of functional correctness rewards in DPO, thereby enhancing optimization effectiveness.

## 4 Experiment

### 4.1 Experiment Setup

**Datasets** Our training dataset is sourced from CodeV [4], consisting of 165k samples obtained by crawling all publicly available Verilog module code on GitHub. However, we observed that a subset of these modules are not syntactically valid, prompting us to filter out such cases and retain a cleaned dataset of 135k samples. The 135k dataset is initially employed to fine-tune the general-purpose code generation model Qwen2.5 Coder Instruct [30], yielding a naive code generator. For every prompt in the 135k dataset, this generator produces 5 candidate module codes, while the corresponding module codes in the dataset serve as reference codes to validate signal correctness during Signal-aware Verification.

**Training Settings** In training, we perform full-parameter fine-tuning for 2 epochs during the SFT, followed by about 1 epoch (7000 steps) of LoRA-based[31] fine-tuning in the Signal-aware DPO. Optimization is carried out using the Adam [32] optimizer with a cosine annealing learning rate schedule, where the initial learning rates are set to 1.0e-5 for SFT and 5.0e-6 for Signal-aware DPO.

Table 1: Main Results on **VerilogEval1.0** and **VerilogEval2.0** Benchmarks

| Type | Model | Size | VerilogEval1.0 (%) | | | | | | VerilogEval2.0 (%) | | | |
|---|---|---|---|---|---|---|---|---|---|---|---|---|
| | | | Machine | | | Human | | | Specification | | Completion | |
| | | | Pass@1 | Pass@5 | Pass@10 | Pass@1 | Pass@5 | Pass@10 | T=0 | T=0.8 | T=0 | T=0.8 |
| Foundation General Model | GPT-3.5 | - | 46.7 | 69.1 | 74.1 | 26.7 | 45.8 | 51.7 | - | - | - | - |
| | GPT-4 | - | 60.0 | 70.6 | 73.5 | 43.5 | 55.8 | 58.9 | 32.0 | 31.7 | 42.3 | 41.6 |
| | GPT-4o | - | 67.7 | 75.5 | 77.2 | 60.1 | 71.4 | 74.5 | 62.5 | 61.4 | 59.0 | 56.1 |
| | Deepseek v3 | 671B | **77.6** | **86.2** | **87.4** | **70.7** | **77.4** | **78.8** | 68.8 | 66.9 | 68.0 | 66.1 |
| General Code Model | Deepseek Coder | 6.7B | **52.2** | 55.4 | 56.8 | **30.2** | 33.9 | 34.9 | 21.7 | 19.5 | 25.0 | **29.3** |
| | CodeQwen1.5 | 7B | 46.5 | 54.9 | 56.4 | 22.5 | 26.1 | 28.0 | 1.9 | 15.0 | 21.8 | 17.9 |
| | Qwen2.5 Coder Instruct | 7B | 50.1 | **66.5** | **70.9** | 22.9 | **36.0** | **39.5** | **23.0** | **22.0** | **30.1** | 25.6 |
| Verilog-Specific Model | RTLCoder | 6.7B | 61.2 | 76.5 | 81.8 | 41.6 | 50.1 | 53.4 | 36.8 | 30.9 | 35.9 | 31.5 |
| | CodeV(Qwen1.5) | 7B | 77.6 | 88.2 | 90.7 | 52.7 | 62.5 | 67.3 | 7.7 | 8.4 | 50.0 | 45.7 |
| | CodeV(Qwen2.5) | 7B | 77.3 | 87.9 | 90.1 | 57.9 | 66.7 | 69.7 | 44.8 | 37.4 | 58.3 | 49.9 |
| | Origen | 7B | 74.1 | 82.4 | 85.7 | 54.4 | 60.1 | 64.2 | 49.3 | 46.8 | 49.3 | 47.2 |
| | VeriSeek | 6.7B | 61.6 | 76.9 | 81.7 | 30.5 | 43.4 | 49.2 | 28.8 | 15.9 | 49.3 | 43.1 |
| | VeriPrefer | 7B | 72.7 | 85.8 | - | 49.7 | 62.3 | - | - | - | 55.7 | 51.3 |
| | QiMeng-SALV (Ours) | 7B | **81.4** | **88.6** | **90.8** | **60.4** | **68.6** | **71.2** | **57.1** | **56.2** | **62.2** | **58.8** |

The training configuration employs a global batch size of 64 and a maximum sequence length of 2048 tokens.

**Metric**    Following previous work [11], the pass@k metric is employed to evaluate model performance, estimating the probability that at least one correct solution is generated within $k$ independent attempts for each problem:

$$pass@k := \mathbb{E}_{\text{problems}} \left[ 1 - \frac{\binom{n-c}{k}}{\binom{n}{k}} \right]. \tag{3}$$

Here, $n \geq k$ denotes the total number of independent solution attempts per problem instance, while $c$ corresponds to the count of functionally correct solutions among these trials.

**Benchmark**    We conduct comprehensive evaluations on the VerilogEval (including VerilogEval1.0 [9] and VerilogEval2.0 [10]) and RTLLM benchmarks (including RTLLM v1.1 [11] and RTLLM v2.0 [12]).

VerilogEval1.0 is divided into two subtasks: Machine and Human. The Machine subset contains 143 Verilog design problems, with each prompt generated by LLMs; the Human subset comprises 156 Verilog design problems, with each prompt manually crafted. VerilogEval2.0 revisit some limitaions of VerilogEval1.0 [9] and support specification-to-RTL tasks in addition to the original code completion task. Since VerilogEval-machine can be overly descriptive compared to real-world code generation problem, VerilogEval2.0 only evaluate models against VerilogEval-human to highlight the most useful LLM evaluation.

RTLLM v1.1 [11] incorporates 29 distinct Verilog code generation tasks categorized into Arithmetic and Logic domains. Each task specification provides complete interface definitions (module ports) along with detailed functional requirements. RTLLM v2.0 expands from the 29 Verilog designs in RTLLM v1.1 to 50, covering four categories of tasks: Arithmetic, Control, Memory, and Miscellaneous.

Our experimental setup employs a sampling configuration of n=20 generations per prompt. We evaluate the model generated module code using pass@1, pass@5, and pass@10 metrics. The pass@1 metric specifically quantifies solution accuracy on demonstrably solvable problems, reflecting the model's consistency and stability in generating correct implementations. In contrast, pass@10 assesses the model's overall problem-solving capacity by measuring its ability to produce at least one valid solution within twenty attempts, thereby characterizing the breadth of its code generation capability.

Following previous practices [16], we conduct tests at temperature settings of 0.2, 0.5, and 0.8, and report the highest results on VerilogEval1.0 and RTLLM benchmark. According to the VerilogEval2.0 paper [10], VerilogEval2.0 only reports pass@1 results under low-temperature settings (T=0.0, top_p=0.01, n=1) and high-temperature settings (T=0.8, top_p=0.95, n=20) using nucleus sampling [33]. Notably, pass@5 and pass@10 scores are excluded from the evaluation.

**Baseline Methods**    In our experimental evaluation, we conduct a comprehensive comparison between our proposed method and several baseline approaches, categorized into three groups: 1)

Table 2: Main Results on **RTLLM v1.1** and **RTLLM v2.0** Benchmarks

| Type | Model | Size | RTLLM v1.1 (%) | | | RTLLM v2.0 (%) | | |
|---|---|---|---|---|---|---|---|---|
| | | | Pass@1 | Pass@5 | Pass@10 | Pass@1 | Pass@5 | Pass@10 |
| Foundation General Model | GPT-3.5 | - | 28.3 | 36.9 | 41.4 | 34.4 | 49.8 | 52.1 |
| | GPT-4o | - | 41.7 | 65.9 | - | 56.5 | 70.3 | **75.2** |
| | Deepseek v3 | 671B | **62.0** | **72.0** | **72.4** | **59.1** | **71.5** | 73.3 |
| General Code Model | Deepseek Coder | 6.7B | 23.1 | 29.3 | 34.5 | 26.5 | 36.3 | 42.7 |
| | CodeQwen1.5 | 7B | 28.8 | 38.8 | 43.3 | 25.8 | 29.0 | - |
| | Qwen2.5 Coder Instruct | 7B | **30.1** | **49.2** | **55.9** | **33.2** | **52.5** | **57.7** |
| Verilog-Specific Model | RTLCoder | 6.7B | 35.8 | 40.3 | 43.1 | 43.5 | 48.1 | - |
| | CodeV(Qwen1.5) | 7B | 36.6 | 53.3 | 61.3 | 48.1 | 56.9 | - |
| | CodeV(Qwen2.5) | 7B | 39.3 | 63.5 | 74.2 | 41.0 | 60.1 | 68.1 |
| | Origen | 7B | 50.6 | 68.3 | 74.3 | 50.9 | 60.9 | 64.0 |
| | VeriSeek | 6.7B | 29.3 | 47.1 | 53.1 | 31.9 | 54.2 | 52.0 |
| | VeriPrefer | 7B | 53.2 | 67.7 | - | 52.4 | 66.4 | - |
| | QiMeng-SALV (Ours) | 7B | **62.6** | **75.1** | **81.1** | **62.0** | **71.7** | **76.0** |

General-purpose Foundation Models: GPT-3.5, GPT-4o, GPT-4 [34], and DeepSeek-v3 [13], which demonstrate broad capabilities across diverse domains. 2) General Code Models: CodeQwen1.5 [35], Qwen2.5 Coder Instruct [30] and Deepseek Coder [36], possessing strong coding proficiency but lacking Verilog-specific optimization. 3) Domain-Specialized Verilog Models: Including RTL Coder [16], CodeV [4], Origen [17], VeriSeek [7] and VeriPrefer [28] which employ module-level rewards in reinforcement learning.

## 4.2 Main Results

Table 1 and Table 2 comprehensively compare the performance of our QiMeng-SALV method against baseline models across the VerilogEval [9, 10] and RTLLM [11, 12] benchmarks. To establish a more rigorous evaluation framework, we re-implemented CodeV using Qwen2.5 Coder Instruct as its foundation model, replacing its original CodeQwen1.5 which demonstrated inferior performance. This architectural alignment with our base model eliminates potential confounding factors arising from fundamental model discrepancies.

Comparison results in Table 1 and Table 2 show that QiMeng-SALV establishes new state-of-the-art results across both benchmarks in the open-source domain. As shown in Table 1, in VerilogEval, QiMeng-SALV achieves leading performance among open-source solutions on both specification understanding and code completion tasks, attaining 81.4 pass@1 on Machine and 60.4 pass@1 on Human in VerilogEval 1.0, as well as 57.1 (T = 0) on Specification and 62.2 (T = 0) on Completion in VerilogEval2.0. Its performance is comparable to GPT-4o on the VerilogEval1.0 Human benchmark and even surpasses Deepseek v3 and GPT-4o on the VerilogEval1.0 Machine.

As shown in Table 2, QiMeng-SALV achieves a remarkable 62.6% functional pass@1 accuracy on the RTLLM v1.1 benchmark and 62.0% on RTLLM v2.0 with merely 7B parameters, significantly exceeding all existing open-source alternatives and rivaling the performance of DeepSeek-v3, a 671B parameter model. Impressively, its functional pass@10 accuracy reaches 81.1% on RTLLM v1.1, surpassing DeepSeek-v3's 72.4%.

Remarkably, when compared to CodeV (Qwen2.5), trained on identical datasets with the same base model, QiMeng-SALV shows substantial improvements: about 59.7% improvement pass@1 accuracy on RTLLM v1.1, and 27.4%(T=0)/50.2%(T=0.8) improvement on VerilogEval2.0's specification subtasks. These findings underscore that leveraging RL to learn from correct signal-level implementations yields substantially stronger performance compared to models trained exclusively via supervised fine-tuning.

Furthermore, when compared with VeriPrefer, which employs module-level rewards in reinforcement learning, our method achieves substantial improvements across all evaluation metrics. This highlights the advantage of fine-grained supervision: signal-level rewards allow the model to effectively learn from partially correct samples that are often disregarded under coarser reward schemes. Consequently, signal-level reinforcement learning enables the generation of more robust and correct Verilog code.

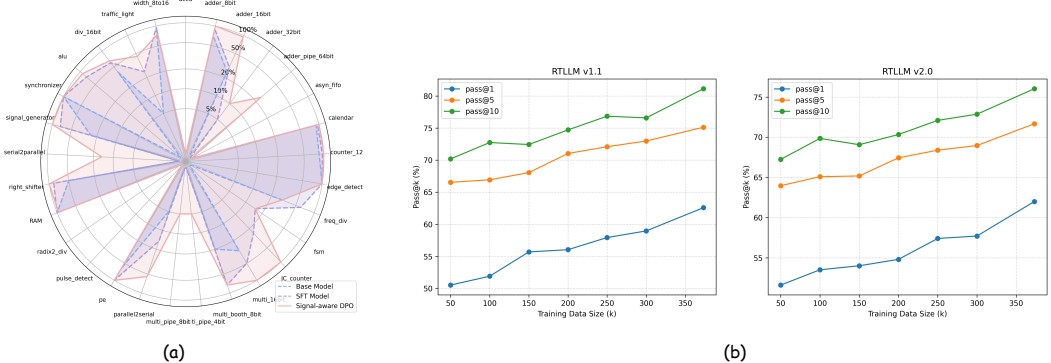

Figure 3: (a) Pass rate performance comparison between different training stages across different tasks on RTLLM v1.1 benchmark. (b) Signal-aware DPO training datasets scaling on RTLLM benchmark.

## 4.3 Ablation Study

In this section, we present a systematic ablation study to evaluate our proposed QiMeng-SALV. Given that RTLLM contains more complex problems than VerilogEval, making it more suitable for revealing nuanced performance differences among models, we focus our ablation analysis on the RTLLM benchmark. Our investigation spans two key dimensions: 1). configuration variations of Signal-aware DPO training, and 2). different training stage implementations of QiMeng-SALV.

**Ablation of configuration variations of signal-aware DPO**    Unlike standard DPO, which is limited to processing complete correct modules and must consider all response tokens, our signal-aware DPO can not only handle complete correct modules but also extract functional reward in partial correct modules containing correct signal implementations. Therefore, we design three different options to evaluate the efficiency of our method: 1) Complete correct datasets (263k): all of preferred sample in DPO training data are complete correct modules. 2) Partial correct datasets (110k): all of preferred sample in DPO training data are partial correct modules containing both correct and incorrect signals. 3) Filter incorrect signals: determines whether erroneous signals in preferred samples participate in DPO computation. As evidenced by the results in Table 3, standard DPO trained exclusively on complete correct datasets achieves superior performance compared to training on partially correct datasets. Interestingly, combining both complete and partially correct datasets during training leads to degraded performance relative to using only complete correct datasets. This degradation likely stems from the presence of incorrect signal implementations in partially correct module, which introduce noise and hinder the model's ability to discern meaningful reward signals. By contrast, our proposed signal-aware DPO method, which actively filters out incorrect signal implementations, substantially improves performance in both scenarios: training on partially correct datasets alone and training on mixed datasets. These findings highlight two key advantages of signal-aware DPO: (1) its ability to leverage a broader spectrum of training data, and (2) its robustness against noisy signal implementations through effective filtering.

**Ablation of different training stage implementations**    To systematically analyze the performance gains of the model across different training phases, we conducted comprehensive evaluations at each stage. The results, as presented in the Table 4, indicate that SFT yields a substantial improvement of approximately 15 percentage points in all three accuracy dimensions. Subsequent signal-aware DPO training delivers additional gains of 10-14 percentage points. Figure 3 (a) illustrates the pass rate of the model on the RTLLM v1.1 benchmark tasks across different training stages. The base code model not only answered fewer questions correctly but also exhibited low accuracy for the questions it could answer. After the SFT training stage, the model could correctly answer a broader range of questions. Following signal-aware DPO training, the model further improved its accuracy on questions it could answer while also correctly answering previously unsolvable questions. These findings substantiate that signal-aware DPO effectively utilizes functional correctness feedback signals to drive additional performance improvements beyond what is achieved through SFT alone.

Table 3: Ablation Different Settings on **RTLLM** Benchmark

| Complete Correct Datasets | Partial Correct Datasets | Filter Incorrect Signals | RTLLM v1.1 (%) | | | RTLLM v2.0 (%) | | |
|:---:|:---:|:---:|:---:|:---:|:---:|:---:|:---:|:---:|
| | | | Pass@1 | Pass@5 | Pass@10 | Pass@1 | Pass@5 | Pass@10 |
| ✓ | | | 57.4 | 74.0 | 78.3 | 57.1 | 70.4 | 73.4 |
| | ✓ | | 56.0 | 67.9 | 71.6 | 54.7 | 66.0 | 69.2 |
| ✓ | ✓ | | 55.3 | 73.1 | 77.1 | 55.9 | 69.8 | 73.3 |
| | ✓ | ✓ | 56.6 | 71.0 | 76.4 | 54.1 | 68.4 | 73.1 |
| ✓ | ✓ | ✓ | **62.6** | **75.1** | **81.1** | **62.0** | **71.7** | **76.0** |

Table 4: Ablation Different Training Stage on **RTLLM** Benchmark

| Model | RTLLM v1.1 (%) | | | RTLLM v2.0 (%) | | |
|:---|:---:|:---:|:---:|:---:|:---:|:---:|
| | Pass@1 | Pass@5 | Pass@10 | Pass@1 | Pass@5 | Pass@10 |
| Base Model(Qwen2.5) | 30.1 | 49.2 | 55.9 | 33.2 | 52.5 | 57.7 |
| + SFT | 48.3 | 65.2 | 69.8 | 50.4 | 64.3 | 68.7 |
| + Signal-aware DPO | **62.6** | **75.1** | **81.1** | **62.0** | **71.7** | **76.0** |

**Scaling on Signal-aware DPO training datasets**    To examine the influence of training dataset size on Signal-aware DPO's efficacy, we conducted experiments by training models with preference datasets of different scales. Figure 3 (b) reveals a consistent upward trend in pass@1, pass@5, and pass@10 metrics as the dataset size grows. These results indicate that incorporating more meaningful functional rewards during reinforcement learning training significantly contributes to improved model performance.

## 4.4   Runtime Analysis

We measured the average runtime of simulation and AST parsing over all Verilog samples in training data. On average, simulation takes 0.0391 seconds and AST parsing takes 0.0957 seconds per sample. By utilizing 60 CPU cores for parallel processing, we reduce the per-sample average simulation time to 0.65 milliseconds and AST parsing time to 1.59 milliseconds. These results show that the overhead of simulation and AST parsing is negligible in practice.

## 5   Conclusion

In this paper, we propose QiMeng-SALV, a novel reinforcement learning method for training Verilog code generation, which learns correct signal implementations from erroneous modules, thereby increasing functional correctness rewards and improving capability. Experiments show that our model achieves state-of-the-art performance on VerilogEval and RTLLM, significantly outperforming other open-source models of comparable scale. On RTLLM v1.1, it reaches pass@1 62.6% and pass@5 75.1%, matching the performance of DeepSeek-v3 (671B parameters) with only 7B parameters. QiMeng-SALV shifts the training paradigm for Verilog code generation from the module level to the signal level.

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

# Supplementary Material

## A   Experiment Compute resources

For the SFT phase, we execute full-parameter optimization across 2 training epochs utilizing a cluster of 4 NVIDIA A100-80GB SMX GPUs, with the complete training procedure consuming approximately 20 hours. Subsequently, during the Signal-aware DPO stage, we implement LoRA-based fine-tuning over 7000 steps on an array of 8 NVIDIA A100-40GB GPUs, resulting in a total training duration of roughly 15 hours.

## B   Societal Impacts

Our research delivers significant positive societal impact by substantially improving the functional correctness of automatically generated Verilog code. This advancement enhances productivity in circuit design workflows, accelerates development cycles, and provides industry practitioners with a reliable assistive tool. However, we acknowledge potential negative implications in academic settings. The model's capabilities could be misused by students to complete Verilog programming assignments or examinations, potentially facilitating academic dishonesty. We emphasize the importance of developing appropriate usage guidelines and detection mechanisms to mitigate such risks while preserving the technology's beneficial applications.

## C   Limitation

QiMeng-SALV determines the functional correctness of the generated module by producing random input signals and comparing the output signals between the generated module and the reference module. If the reference module in the training dataset itself is incorrect, it may lead to erroneous judgments of functional correctness, necessitating a relatively high-quality training dataset. In this paper, we do not discuss how to obtain a high-quality dataset, as this is beyond our scope. In our experiment, we use the CodeV dataset, a high-quality dataset obtained by crawling all publicly available Verilog module code on GitHub. We also perform data cleaning to ensure the reference modules are syntactically correct and compilable.

