# OpenReview forum: "QiMeng-SALV: Signal-Aware Learning for Verilog Code Generation"
_NeurIPS.cc/2025/Conference — NeurIPS 2025 poster_

### Official Review · Reviewer_EgJ3 · 2025-06-29

**Clarity:** 3
**Significance:** 3
**Originality:** 3
**Rating:** 4
**Confidence:** 3

**Summary:**

This paper proposes a signal-aware learning framework (SAL-V) for Verilog code generation, using AST-based extraction of functionally correct code segments to provide fine-grained RL rewards. The approach addresses limitations of module-level rewards and achieves state-of-the-art results on multiple benchmarks, showing strong performance even with smaller models.

**Questions:**

For Table 1, why does the VeriSeek method achieve such a notably low score on the Specification task?

**Ethical Concerns:**

["NO or VERY MINOR ethics concerns only"]

**Final Justification:**

I maintain my raw score.

**Limitations:**

yes

**Quality:**

3

**Strengths And Weaknesses:**

Strength:

- Employing abstract syntax tree (AST) analysis to identify signal-aware code segments is a novel approach for data extraction, and it is well-suited to Verilog code generation. This represents an interesting problem and a promising direction for further exploration.
- The experimental results validate the effectiveness of the proposed method across multiple datasets, demonstrating its robustness and broad applicability.

Weakness:

- A major weakness is the risk of data contamination in the CodeV dataset, as the 165k training samples likely overlap or are similar to those in the test set. This raises concerns about overfitting and whether the reported performance truly reflects generalization rather than memorization.
- There is an inconsistency in GPT-4o’s Pass@1 results on the RTLLM benchmark: this paper reports 33.8, while CodeV-R1 Table 2 reports 56.8. This discrepancy makes it unclear which result is correct and complicates fair comparison. Please confirm which version of RTLLM was used.

- The use of reinforcement learning to improve Verilog generation has already been explored in VeriSeek[1], and leveraging Abstract Syntax Tree-based methods was previously discussed in the “Data is All You Need” [2]. Thus, the novelty of these aspects in this work appears limited.

[1] Wang, Ning, et al. "Large Language Model for Verilog Generation with Golden Code Feedback." arXiv preprint. 2024.

[2] Chang, Kaiyan, et al. "Data is all you need: Finetuning llms for chip design via an automated design-data augmentation framework." DAC. 2024.

---

> ### Author Rebuttal · Authors · 2025-07-30
>
> **Comments on Weaknesses:**
>
> **W1: A major weakness is the risk of data contamination in the CodeV dataset, as the 165k training samples likely overlap or are similar to those in the test set. This raises concerns about overfitting and whether the reported performance truly reflects generalization rather than memorization.**
>
> We appreciate the reviewer’s concern regarding potential data contamination. According to the original CodeV [1] paper, a decontamination procedure was applied to mitigate such risks. Specifically, CodeV uses ROUGE-L, a widely adopted similarity metric, to detect overlap between the training and test sets, and removes any training sample with a ROUGE-L score greater than 0.5 relative to the test set.
>
> In our experiments, we used the same decontaminated dataset and the same base model to train both SAL-V and the CodeV (Qwen2.5) baseline. Despite these identical settings, our method significantly outperforms CodeV (Qwen2.5), which suggests that the performance gain is not due to overfitting or memorization. Rather, it indicates that SAL-V is more effective at learning from the available data and exhibits stronger generalization capabilities.
>
> [1] Zhao Y, Huang D, Li C, et al. CodeV: Empowering LLMs with HDL Generation through Multi-Level Summarization[J]. arXiv preprint arXiv:2407.10424, 2024.
>
> **W2: There is an inconsistency in GPT-4o’s Pass@1 results on the RTLLM benchmark: this paper reports 33.8, while CodeV-R1 Table 2 reports 56.8. This discrepancy makes it unclear which result is correct and complicates fair comparison. Please confirm which version of RTLLM was used.**
>
> Thank you for your valuable feedback. We would like to clarify the source of the discrepancy regarding GPT-4o’s performance on the RTLLM benchmark. The Pass@1 score of 33.8 reported in our paper is based on RTLLM v1.1 sourced from CraftRTL [1]. In contrast, CodeV-R1 reports a score of 56.8 on RTLLM v2.0, which is a newer and different version of the benchmark. For reference, CodeV-R1 also reports a Pass@1 of 41.7 for GPT-4o on RTLLM v1.1, which is more consistent with the 33.8 reported in our paper. It’s also important to note that the CodeV-R1 results were released after the NeurIPS submission deadline, and were therefore not available during the preparation of our paper.
>
> For completeness, we have included our model's performance on RTLLM v2.0 in the appendix. Our method achieves Pass@1 = 56.6, Pass@5 = 68.5, and Pass@10 = 71.7, demonstrating comparable performance to GPT-4o on this more recent benchmark.
>
> [1] Liu M, Tsai Y D, Zhou W, et al. CraftRTL: High-quality synthetic data generation for Verilog code models with correct-by-construction non-textual representations and targeted code repair. arXiv preprint arXiv:2409.12993, 2024.
>
> **W3: The use of reinforcement learning to improve Verilog generation has already been explored in VeriSeek[1], and leveraging Abstract Syntax Tree-based methods was previously discussed in the “Data is All You Need” [2]. Thus, the novelty of these aspects in this work appears limited.**
>
> Thank you for the thoughtful feedback. While RL and ASTs have been used in prior works, our approach differs significantly in both motivation and implementation.
>
> VeriSeek [1] uses structural similarity between the ASTs of generated and reference code as the reward. However, such similarity does not guarantee functional correctness, limiting its practical utility. In contrast, our reward is explicitly grounded in functional correctness: we simulate the generated code, identify which output signals behave correctly, and then use AST-based analysis to trace and extract the exact code segments responsible for those signals. This enables fine-grained, signal-level rewards that directly target correct functionality, which is fundamentally more informative than structural matching.
>
> Similarly, although [2] also uses ASTs, their goal is to translate structural elements of Verilog code into natural language descriptions. Our use of ASTs is entirely different: we perform signal dependency analysis to isolate functionally meaningful code segments for reward calculation. This purpose-driven use of ASTs is novel in the context of reward design for Verilog generation.
>
> [1] Wang, Ning, et al. "Large Language Model for Verilog Generation with Golden Code Feedback." arXiv preprint. 2024.
>
> [2] Chang, Kaiyan, et al. "Data is all you need: Finetuning llms for chip design via an automated design-data augmentation framework." DAC. 2024.
>
> **Reply to Questions:**
>
> **Q1: For Table 1, why does the VeriSeek method achieve such a notably low score on the Specification task?**
>
> Thank you for the valuable feedback. The poor performance of VeriSeek on the Specification task is primarily due to its inability to generate correct module headers when they are not provided in the prompt. Unlike other tasks, the Specification task omits the module header, requiring the model to synthesize it purely from the description. VeriSeek often fails in this aspect, likely because it relies on AST similarity as the RL reward signal, rather than functional or syntactic correctness. Without explicitly optimizing for executable or compilable outputs, the model struggles to produce structurally valid modules, especially when critical elements like headers must be generated from scratch.
>
> Moreover, VeriSeek tends to use an outdated Verilog style where input/output declarations and bitwidths are placed inside the module body. In contrast, VerilogEval 1.0 and 2.0 adopt the modern convention of declaring them directly in the module header. This style mismatch often leads to compilation or execution failures during evaluation.
>
> To verify this, we convert the VeriSeek’s generated module headers to modern style. After the fix, its performance improved (e.g., pass@1: 28.8 for T=0 and 15.9 for T=0.8), but still lagged significantly behind other models. This confirms that the lack of module header generation capability is a key limitation of VeriSeek in this setting.

---

### Official Review · Reviewer_2wdY · 2025-07-01

**Clarity:** 3
**Significance:** 3
**Originality:** 3
**Rating:** 4
**Confidence:** 4

**Summary:**

This paper proposes SAL-V, a novel signal-aware learning approach for Verilog code generation. The key idea is to leverage functionally correct output signal code segments to optimize RL training. The method addresses the challenge of insufficient functional rewards in LLM-based Verilog generation by identifying and utilizing correct signal-level implementations from partially incorrect modules. It introduces a signal-aware DPO that focuses on correct signal-level code segments. The approach achieves state-of-the-art performance, demonstrating a significant advancement in the field.

**Questions:**

Please refer to the weaknesses.

**Ethical Concerns:**

["NO or VERY MINOR ethics concerns only"]

**Final Justification:**

I have read the rebuttal and will maintain my score.

**Limitations:**

yes

**Quality:**

3

**Strengths And Weaknesses:**

Strengths:
1. The paper introduces a new paradigm for Verilog code generation, shifting from module-level to fine-grained signal-level optimization. This is a crucial innovation that addresses a fundamental limitation of existing RL-based methods, which struggle with sparse functional rewards.


2. SAL-V's ability to extract meaningful functional rewards from erroneous modules by verifying and isolating correct signal implementations is highly impactful. The use of AST for signal-aware code extraction is a clever technical contribution.


3. The experimental results are compelling, showing that SAL-V achieves competitive and even superior performance compared to much larger models (e.g., matching DeepSeek-V3 671B with a 7B model) on challenging benchmarks like VerilogEval2.0 and RTLLM.


Weaknesses:
1. While the proposed signal-aware learning is highly effective, it would be beneficial to see the methodology validated on large-scale on-policy RL. The potential of off-policy RL for very large-scale training might be limited, and demonstrating the applicability and scalability of SAL-V within an on-policy framework would further strengthen the paper's claims and implications for future research.

---

> ### Author Rebuttal · Authors · 2025-07-30
>
> **Comments on Weaknesses:**
>
> **W1: While the proposed signal-aware learning is highly effective, it would be beneficial to see the methodology validated on large-scale on-policy RL. The potential of off-policy RL for very large-scale training might be limited, and demonstrating the applicability and scalability of SAL-V within an on-policy framework would further strengthen the paper's claims and implications for future research.**
>
> We appreciate the reviewer’s suggestion. We initially chose DPO for its sampling efficiency and low computational cost and it allows us to pre-sample generated code before training, making it well-suited for resource-efficient RL.
>
> To evaluate the scalability of our approach in an online RL setting, we implemented SAL-V using GRPO [1], an improved online RL algorithm built upon PPO [2].
>
> | Reward Type |  Model |｜ |   |   RTLLM   |  | ｜|   |   Machine  | |｜|   |  Human   | |
> |:---:|:---:|:---:|:---:|:---:|:---:|---:|:---:|:---:|:---:|-----:|:----:|:-----:|:---:|
> |   |   | ｜ | Pass@1 | Pass@5 | Pass@10 | ｜ | Pass@1 | Pass@5 | Pass@10 | ｜| Pass@1 | Pass@5 | Pass@10 |
> | Signal-level | SAL-V DPO   |  ｜| 61.5  |  **75.9**  |  **80.3**   |  ｜ |   79.8 |  88.0  |  **90.3**   |  ｜|  60.1 |  67.6  |**70.4** |
> | Module-level | Normal GRPO |  ｜|57.7  |  72.6  |  78.4   |  ｜|  82.6  |  86.0  |  86.8   |｜|  60.9 |  65.2  |66.5|
> | Signal-level | SAL-V GRPO  |  ｜ |    **61.5**  |  75.1  |  80.1   |  ｜| **84.7**  |  **88.2**  |  88.9   |｜ | **64.5**   |  **68.4**  |69.3|
>
> As illustrated by the results, SAL-V combined with GRPO (online RL) achieves better performance than with DPO (offline RL) on both VerilogEval-Machine and VerilogEval-Human, and performs comparably on RTLLM. This indicates that SAL-V is compatible with both offline and online RL algorithms, showcasing strong generalizability across training paradigms.
>
> In addition, using signal-level rewards with GRPO consistently outperforms module-level reward schemes on all benchmarks. This highlights the benefit of providing more detailed and targeted feedback during training, allowing the model to better leverage partially correct outputs and improve overall Verilog code generation performance.
>
> [1] Shao Z, Wang P, Zhu Q, et al. Deepseekmath: Pushing the limits of mathematical reasoning in open language models[J]. arXiv preprint arXiv:2402.03300, 2024.
>
> [2] Schulman J, Wolski F, Dhariwal P, et al. Proximal policy optimization algorithms[J]. arXiv preprint

---

### Official Review · Reviewer_voDh · 2025-07-02

**Clarity:** 2
**Significance:** 3
**Originality:** 2
**Rating:** 4
**Confidence:** 4

**Summary:**

This paper introduces SAL-V, a reinforcement learning (RL) approach to Verilog code generation, emphasizing fine-grained signal-aware rewards and Direct Preference Optimization (DPO). Unlike conventional methods that use module-level rewards, SAL-V employs signal-level correctness rewards extracted through Yosys AST parsing and dependency analysis, guiding the generation of Verilog code. Experimental evaluation demonstrates significant performance improvements on established benchmarks, surpassing several strong baselines.

**Questions:**

1.	Could you clarify which base models were used to train SAL-V and how they were configured (e.g., frozen layers, RL steps, DPO parameters)?
2.	Why were simpler RL strategies like module-level reward (e.g., from Insights from Verification) not included for comparison?
3.	How does your signal extraction pipeline handle ambiguous or non-standard Verilog code?
4.	Is the Yosys-based extraction scalable to large, complex modules?
5.	Could you provide a brief overview of the distribution of your signal-aware dataset—how many signals, what kinds, and how are preference pairs balanced?

**Ethical Concerns:**

["NO or VERY MINOR ethics concerns only"]

**Final Justification:**

The authors have addressed my concerns and I would like to raise my score from 3 to 4.

**Limitations:**

The authors partially acknowledge limitations, but further elaboration is necessary on failure modes of signal-level extraction, scalability, and comparisons with simpler methods.

**Paper Formatting Concerns:**

no formatting issue.

**Quality:**

2

**Strengths And Weaknesses:**

Strengths:
1. The proposed signal-level correctness reward approach significantly refines traditional module-level RL feedback, potentially offering better learning signals.
2. Evaluations across VerilogEval2.0 and RTLLM benchmarks demonstrate the method’s superior performance compared to strong baselines such as GPT-4o and DeepSeek-v3.
3. DPO provides more stable training compared to PPO, aligning effectively with the structured and fine-grained nature of Verilog code.
4. Detailed ablation studies demonstrate the contributions of various components of the proposed method.

Weaknesses:
1. The baseline models' details, such as their specific architectures and training setups, are not clearly articulated. The paper does not explicitly state or adequately discuss the choice of base models.
2. Presentation Issues: Some typographical issues were observed, including inconsistent decimal alignment in Table 1 for "CodeV(Qwen1.5)" and suboptimal sizing of Figure 2(a), making the framework overview challenging to interpret clearly.
3. Incomplete Related Work Discussion: While several recent works have explored RL-based RTL generation, such as "Insights from Verification: Training a Verilog Generation LLM with Reinforcement Learning with Testbench Feedback," the authors do not provide comparative analysis with these related approaches. Given the lack of theoretical justification and limited experimental settings, omitting such comparisons makes the conclusions regarding the superiority and necessity of the signal-aware method less convincing.

---

> ### Author Rebuttal · Authors · 2025-07-30
>
> **Comments on Weaknesses:**
>
> **W1: The baseline models' details, such as their specific architectures and training setups, are not clearly articulated. The paper does not explicitly state or adequately discuss the choice of base models.**
>
> Thank you for your valuable feedback. We clarify the choice of base model and the associated training setup as follows:
>
> We used Qwen2.5 Coder Instruct as the base model, as our goal is to train a code generation model (not a code completion model like CodeV) that can generate full Verilog modules, even without module headers in the prompt. Qwen2.5 Coder Instruct is well-suited for generating structured programming code and easily transferable to Verilog generation tasks.
>
> We first fine-tuned Qwen2.5 Coder Instruct on a 162k cleaned CodeV dataset to obtain an SFT model. During the SFT stage, we performed full fine-tuning for 2 epochs with a learning rate of 1e-5 and batch size of 64.
>
> Using the SFT model, we sampled 5 outputs for each Verilog design and collected 143k preference pairs based on signal correctness: generations with correct signals were treated as preferred samples, and incorrect ones as dispreferred. We then applied our signal-aware DPO method to fine-tune the SFT model using LoRA. Specifically, we applied LoRA to the linear projection matrices ($W_q, W_k, W_v, W_{out}$) of each transformer layer. The DPO fine-tuning used a batch size of 64, a learning rate of 5e-6, and was trained for 3 epochs (6,756 steps in total).
>
> **W2: Presentation Issues: Some typographical issues were observed.**
>
> Thank you for pointing out these presentation issues. We will fix these issues in the revised paper.
>
> **W3: Incomplete Related Work Discussion: While several recent works have explored RL-based RTL generation, such as "Insights from Verification: Training a Verilog Generation LLM with Reinforcement Learning with Testbench Feedback," the authors do not provide comparative analysis with these related approaches.**
>
> Thank you for your insightful feedback. While VeriPrefer also leverages functional rewards for training, our method differs in several key aspects:
>
> 1. Reward Granularity: VeriPrefer uses module-level rewards, meaning only entirely correct modules receive a reward. In contrast, our method introduces signal-level rewards, a more fine-grained and novel approach. This allows partially correct modules to still contribute useful feedback during training if they contain correctly implemented signals. As a result, our method provides denser and more informative rewards, which leads to better RL performance.
>
> 2. Correctness Evaluation: Our approach evaluates signal correctness by executing both the testbench and golden reference code with randomly generated inputs and comparing their output signals. VeriPrefer, on the other hand, relies on general-purpose testbench generation without access to reference code, using commercial LLMs. This may lead to unreliable or noisy feedback during training.
>
> 3. Training Paradigm: We propose a signal-aware DPO method specifically tailored for Verilog code generation, enabling fine-grained RL training at the signal level. VeriPrefer applies a standard DPO pipeline without such signal-level supervision.
>
> These distinctions enable our method to provide a denser and more reliable reward signal, making it better suited for improving model performance in the context of Verilog code generation. We will incorporate this comparative analysis into the revised version of the paper.
>
> **Reply to Questions:**
>
> **Q1: Could you clarify which base models were used to train SAL-V and how they were configured (e.g., frozen layers, RL steps, DPO parameters)?**
>
> Thank you for the thoughtful question. Please refer to W1.
>
> **Q2: Why were simpler RL strategies like module-level reward (e.g., from Insights from Verification) not included for comparison?**
>
> Thank you for this insightful feedback. Our ablation study actually includes a comparison between module-level and signal-level rewards on RTLLM v1.1 in Table3 in our paper.
> |Reward Type|｜||Syntax(%) | |｜| | Func (%)||
> |:-:|:-:|:-:|:-:|:-:|:-:|:-:|:-:|:-:|
> ||｜|Pass@1| Pass@5 | Pass@10 | ｜|Pass@1 | Pass@5 |Pass@10|
> |Module-level|｜|84.6|90.1|92.1|｜|52.9|70.6|76.2|
> |Signal-level|｜|**91.8**| **96.2**| **96.5**|｜|**61.5**|**75.9**|**80.3**|
>
> We also provide the comparison between SAL-V and VeriPrefer on RTLLM v1.1, VerilogEval Machine and Human benchmarks:
>
> |Reward Type|Model|｜|RTLLM| |｜| Machine | |｜| Human| |
> |-|-|-|-|-|-|-|-|-|-|-|
> | | |｜|pass@1|pass@5|｜|pass@1|pass@5|｜|pass@1|pass@5|
> |Module-level| VeriPrefer |｜| 53.2   |67.7|｜|72.7|85.8|｜|49.7|62.3|
> |Signal-level|SAL-V|｜|**61.5**|**75.9**|｜|**79.8**|**88.0**|｜|**60.1**|**67.6**|
>
> As the results show, the signal-level reward method consistently outperforms the module-level approach across all evaluation metrics. This demonstrates the advantage of fine-grained supervision: signal-level rewards enable the model to effectively learn from partially correct samples, which are often ignored in coarser reward schemes. As a result, signal-level RL leads to more robust and performant Verilog code generation.
>
> **Q3: How does your signal extraction pipeline handle ambiguous or non-standard Verilog code?**
>
> Thank you for your valuable feedback. In SAL-V, our signal extraction pipeline is designed to be robust by operating only on code samples that produce valid execution results. This ensures that the code is syntactically correct and compilable, allowing us to reliably construct ASTs and extract signal-relevant code segments through a rule-based analysis. While ambiguous or non-standard Verilog code may occur in model outputs, such samples typically fail during simulation or compilation and are therefore excluded from the signal extraction pipeline.
>
> To assess the reliability of our extraction process, we conducted a compilation-based evaluation on all the generated samples from the signal-aware DPO stage that contain at least one correct signal. After extracting the relevant code segments of correct signals, we tested their compilability using iverilog [1]. The compilation success rate was 94.9%, indicating that the extracted code is not only minimal and complete, but also syntactically valid and compilable.
> These results confirm the robustness and reliability of our AST-based signal extraction pipeline.
>
> [1] S. Williams, “Icarus verilog,” https://github.com/steveicarus/iverilog, 2000.
>
> **Q4: Is the Yosys-based extraction scalable to large, complex modules?**
>
> Yes, our Yosys-based signal extraction method is scalable to large and complex Verilog modules. Such modules are typically composed of hierarchically nested submodules. Our AST-based analysis supports recursive traversal across module instantiations, allowing signal dependencies to be accurately traced through multiple layers of the design hierarchy.
> Let us illustrate this with an example:
> ```
> module lab1_3 (a, b, c, aluctr, d, e);
>    input [1:0] aluctr;
>    input [3:0] a, b, c;
>     output [3:0] d;
>     output [3:0] e;
>     wire [2:0] sel;
>     assign sel = {aluctr[1], aluctr[0], aluctr[0]};
>     mux7 MA (.a(d), .b(e), .c(c), .sel(sel), .out(e[0]));
>     mux7 MB (.a(b), .b(~a), .c(a), .sel(sel), .out(e[1]));
>     mux7 MC (.a(b), .b(a), .c(~a), .sel(sel), .out(e[2]));
>     mux7 MD (.a(a), .b(b), .c(a^b), .sel(sel), .out(e[3]));
>     ALU adder0(a,b,c,aluctr,d);
> endmodule
> module mux7(a, b, c, sel, out);
>    input [3:0] a, b, c;
>    input [2:0] sel;
>     output out;
>    assign out = sel[2] ? (sel[1] ? (sel[0] ? c[3] : c[2]) : (sel[0] ? c[1] : c[0])) :
>                        (sel[1] ? (sel[0] ? b[3] : b[2]) : (sel[0] ? b[1] : b[0]));
> endmodule
> module ALU(a, b, cin, aluc, sum);
> ....
> endmodule
> ```
> Suppose we aim to extract the code relevant to signal `d` in the top-level module `lab1_3`. Through AST analysis, we determine that signal `d` is connected to the signal `out` of a submodule `mux7`, which is instantiated inside the top module. Within `mux7`, `out` depends on signals `a`, `b`, `c`, and `sel`. These internal signals correspond to `a`, `b`, `c`, and `aluctr` in module `lab1_3`. As a result, we determine that `d` depends only on `a`, `b`, `c`, and `aluctr`, and is independent of other signals such as `sel` and `e` in module `lab1_3`.
>
> This recursive dependency tracing enables us to extract only the minimal and functionally complete subset of code relevant to signal `d`, even in the presence of deep modular hierarchies. Thus, the methodology remains robust and scalable for large and complex designs.
>
> **Q5: Could you provide a brief overview of the distribution of your signal-aware dataset—how many signals, what kinds, and how are preference pairs balanced?**
>
> Yes, thank you for your valuable feedback. Below is a brief overview of our signal-aware dataset and the construction of preference pairs. We will also include these details in the revised version of the paper for clarity and completeness.
>
> Among the 165k Verilog design problems in CodeV, 124k are simulation-valid. Of these, 71k (57.8%) are single-output signal tasks, while 49k (39.2%) involve multiple output signals, highlighting the prevalence of multi-signal designs.
>
> For each problem, we sampled 5 generations and constructed preference pairs by treating generations containing at least one correct signal as preferred, and those without any correct signals as dispreferred. After removing duplicates, we obtained 143k preference pairs for training our signal-aware DPO model.
>
> Among them, 106k pairs have preferred samples that are entirely correct (no incorrect signals). We denote this as the Complete Correct Dataset. The remaining 43k are Partial Correct Dataset pairs, where the preferred sample contains both correct and incorrect signals.
> Our ablation study in section 4.3 further shows that the Partial Correct Dataset contributes valuable reward signals under our signal-aware training framework, enhancing model performance.

---

> > ### Comment · Reviewer_voDh · 2025-08-06
> >
> > Thank you for the detailed rebuttal. My concerns have been addressed.

---

> > > ### Author Response · Authors · 2025-08-09
> > >
> > > Dear Reviewer,
> > >
> > > We are pleased to learn that our rebuttal and the corresponding revisions have addressed your concerns. We truly appreciate the time and thoughtful feedback you have provided throughout this review process. With your concerns now resolved, we sincerely hope that our work will earn your support in the final evaluation.
> > >
> > > Best regards,
> > > The Authors

---

> ### Author Response · Authors · 2025-08-06
>
> Dear Reviewer,
>
> We sincerely thank you for taking the time to review our paper. As the discussion phase draws to a close, we would like to kindly ask whether our response has adequately addressed your main concerns. We would greatly appreciate it if you could take a moment to review our replies and share your thoughts in the discussion, indicating whether your concerns have been resolved. Your feedback would be greatly appreciated in helping ensure a fair and thorough evaluation of our work.
>
> Thank you again for your time and thoughtful contributions.

---

### Official Review · Reviewer_q7XC · 2025-07-03

**Clarity:** 3
**Significance:** 3
**Originality:** 3
**Rating:** 4
**Confidence:** 3

**Summary:**

A key challenge in RL-based code generation for hardware is the lack of good rewards, existing approaches rely on structural similarity between generated and reference code, which doesn’t always reflect functional correctness. SAL-V tackles this by shifting the optimization focus from the entire module level to the level of individual output signals. they extract these correct signal segments by checking signal correctness via simulation and identifying the corresponding AST subtrees. Their approach leads to stronger and more targeted learning signals during RL

**Questions:**

Can you report training cost and overhead introduced by simulation + AST parsing?
Could you compare signal-wise rewards to other fine-grained alternatives?
Is SAL-V effective when used with other RL algorithms beyond DPO?
Can you include qualitative examples and error analysis of signal-wise segment extraction?
How does SAL-V perform when signal outputs are interdependent?
How reliable is AST extraction when the generated code is invalid or malformed?

**Ethical Concerns:**

["NO or VERY MINOR ethics concerns only"]

**Final Justification:**

I am maintaining my score of 4 (Borderline Accept). The core contribution of formulating RL rewards at the signal level in Verilog code generation is both novel and technically well-grounded. The authors show that the signal independence assumption is valid empirically which makes a strong case for the paper.

**Limitations:**

yes in appendix.

**Quality:**

2

**Strengths And Weaknesses:**

- Signal-level reward formulation is novel and well-justified,  shifts RL optimization from module-level to signal-level using functionally verifiable outputs, enabling learning from partially correct generations.
- Seamless integration into Direct Preference Optimization training. The method doesn’t require architecture changes, making it easy to plug into existing SFT+DPO pipelines for code generation.
- Assumption of signal independence is untested, no experiments probe scenarios with dependent or correlated signals, which are common in real designs, is it a safe assumption? All evaluations are conducted on curated, synthetic datasets.
- No runtime analysis is presented to assess overhead of simulation and AST parsing.
- Comparison with alternative reward schemes?

The paper presents a creative, well-motivated, and technically sound method for improving reward fidelity in Verilog code generation. However, the paper would benefit from more thorough analysis in several areas—robustness to signal dependencies, runtime costs, qualitative behavior, and evaluation beyond curated benchmarks. These limitations suggest that while the core contribution is strong, its generalization and practical deployment may still require further work.

---

> ### Author Rebuttal · Authors · 2025-07-30
>
> **Comments on Weaknesses:**
>
> **W1: Assumption of signal independence is untested, no experiments probe scenarios with dependent or correlated signals, which are common in real designs, is it a safe assumption? All evaluations are conducted on curated, synthetic datasets.**
>
> Thank you for the thoughtful feedback. We carefully examined the assumption both conceptually and empirically and our analysis shows that the assumption of signal independence is safe.
>
> In our setup, we consider an output signal to be independent of another if its minimal and complete implementation does not rely on the other output signal. Even if two output signals depend on a shared intermediate signal, as long as one does not directly or indirectly consume the value of the other, we treat them as independent. This assumption allows us to isolate and reward correct output signal code segments without being affected by incorrect ones.
>
> To evaluate the safety of this assumption in practice, we conducted a large-scale analysis using AST parsing on all training samples that contain both correct and incorrect output signals. We found that only 4.8% (1,760 out of about 36k) of these samples exhibited cases where a correct signal directly or indirectly depended on an incorrect one. Given this low occurrence, we filtered out such samples during training.
>
> Thus, both by design and through empirical analysis, we conclude that the signal independence assumption is reasonable and does not compromise the validity or effectiveness of our approach. We emphasize that signal-aware code extraction is applied only during training to improve learning. During inference, we apply no additional processing and use the same benchmarks as prior work.
>
> **W2: No runtime analysis is presented to assess overhead of simulation and AST parsing.**
>
> Thank you for raising this point. Our experiment shows that the overhead introduced by simulation and AST parsing is negligible.
> We measured the average runtime of simulation and AST parsing over all Verilog samples in training data. On average, simulation takes 0.0391 seconds and AST parsing takes 0.0957 seconds per sample. By utilizing 60 CPU cores for parallel processing, we reduce the per-sample average simulation time to 0.65 milliseconds and AST parsing time to 1.59 milliseconds. These results show that the overhead of simulation and AST parsing is negligible in practice.
>
> **W3: Comparison with alternative reward schemes?**
>
> Thank you for this insightful feedback. To the best of our knowledge, SAL-V is the first method to introduce a fine-grained, signal-level reward mechanism for Verilog code generation. Thus, we conduct a thorough comparison with the most relevant alternative: standard module-level reward used in conventional DPO on RTLLM v1.1. The result is also presented in Table 3 of the ablation study.
> | Reward Type | ｜  | |Syntax(%) | |｜| | Func (%) | |
> |:--:|:--:|:-:|:--:|:-:|:--:|:-:|:-:|:-:|
> |  | ｜ | Pass@1 | Pass@5 | Pass@10 | ｜|Pass@1 | Pass@5 | Pass@10 |
> | Module-level |｜ | 84.6   | 90.1   | 92.1 | ｜|52.9| 70.6 | 76.2 |
> | Signal-level  | ｜| **91.8** | **96.2** | **96.5**    | ｜| **61.5**| **75.9** | **80.3**|
>
> As the results demonstrate, the signal-level reward method outperforms the module-level approach across all evaluation metrics. This indicates that signal-level rewards offer more fine-grained supervision, allowing the model to learn from partially correct samples and ultimately improving overall performance when such samples are included.
>
> **Reply to Questions:**
>
> **Q1: Can you report training cost and overhead introduced by simulation + AST parsing?**
>
> Thank you for the thoughtful question. For the overhead of simulation and AST parsing, please refer to W2. We would like to clarify that both simulation and AST parsing are performed offline as part of the dataset preprocessing pipeline and do not incur training-time overhead.
> Moreover, modifying the DPO loss for signal-aware reward computation does not introduce additional training cost.
>
> **Q2: Could you compare signal-wise rewards to other fine-grained alternatives?**
>
> Thank you for the thoughtful question. Please refer to W3.
>
> **Q3: Is SAL-V effective when used with other RL algorithms beyond DPO?**
>
> Thank you for the excellent question. To assess the scalability and general applicability of SAL-V, we implemented it in an online RL setting using GRPO [1], an improved version of PPO [2]. The results are summarized in the following table:
>
> | Reward Type |  Model |｜ |   |   RTLLM   |  | ｜|   |   Machine  | |｜|   |  Human   | |
> |:---:|:---:|:---:|:---:|:---:|:---:|---:|:---:|:---:|:---:|-----:|:----:|:-----:|:---:|
> |   |   | ｜ | Pass@1 | Pass@5 | Pass@10 | ｜ | Pass@1 | Pass@5 | Pass@10 | ｜| Pass@1 | Pass@5 | Pass@10 |
> | Signal-level | SAL-V DPO   |  ｜| 61.5  |  **75.9**  |  **80.3**   |  ｜ |   79.8 |  88.0  |  **90.3**   |  ｜|  60.1 |  67.6  |**70.4** |
> | Module-level | Normal GRPO |  ｜|57.7  |  72.6  |  78.4   |  ｜|  82.6  |  86.0  |  86.8   |｜|  60.9 |  65.2  |66.5|
> | Signal-level | SAL-V GRPO  |  ｜ |    **61.5**  |  75.1  |  80.1   |  ｜| **84.7**  |  **88.2**  |  88.9   |｜ | **64.5**   |  **68.4**  |69.3|
>
>
> As shown, online SAL-V (GRPO) outperforms offline SAL-V (DPO) on both VerilogEval-Machine and VerilogEval-Human, while achieving comparable performance on RTLLM. This demonstrates that SAL-V is not only effective in offline settings (e.g., DPO), but also generalizes well to online reinforcement learning frameworks like GRPO.
>
> Moreover, signal-level SAL-V with GRPO consistently outperforms its module-level counterpart across all benchmarks. These results further highlight the benefit of using fine-grained and informative reward signals, which help the model learn more effectively and generalize better across tasks.
>
> [1] Shao Z, Wang P, Zhu Q, et al. Deepseekmath: Pushing the limits of mathematical reasoning in open language models[J]. arXiv preprint arXiv:2402.03300, 2024.
>
> [2] Schulman J, Wolski F, Dhariwal P, et al. Proximal policy optimization algorithms[J]. arXiv preprint
>
> **Q4: Can you include qualitative examples and error analysis of signal-wise segment extraction?**
>
> Yes, for example, consider a training sample involving the design of a controller for a 4-digit 7-segment display. The model generates the following code:
>
> ```
> module anode_controller(
>     input [1:0] refresh_count,
>     output reg [3:0] anode,
>     output reg dp
>     );
>     always @ (*)
>         case(refresh_count)
>         2'b00: begin
>           anode = 4'b1110;
>           dp = 0;
>         end
>         2'b01: begin
>           anode = 4'b1101;
>           dp = 0;
>         end
>         2'b10: begin
>           anode = 4'b1011;
>           dp = 1;
>         end
>         2'b11: begin
>           anode = 4'b0111;
>           dp = 1;
>         end
>         default: anode = 4'b1111;
>         endcase
> endmodule
> ```
> In this case, the output signal anode is correct, while dp is incorrect. In SAL-V, when we extract the code relevant to the anode signal for reward assignment, the extracted minimal code snippet is:
> ```
> module anode_controller(
>     input [1:0] refresh_count,
>     output reg [3:0] anode,
>     );
>     always @ (*)
>         case(refresh_count)
>         2'b00: begin
>           anode = 4'b1110;
>         end
>         2'b01: begin
>           anode = 4'b1101;
>         end
>         2'b10: begin
>           anode = 4'b1011;
>         end
>         2'b11: begin
>           anode = 4'b0111;
>         end
>         default: anode = 4'b1111;
>         endcase
> endmodule
> ```
> This extracted code snippet contains only the minimal set of tokens required to generate the correct anode signal and doesn't contain any incorrect output signal. During signal-aware DPO training, only the tokens corresponding to this relevant subset are included in the loss computation, allowing the model to focus learning on the functionally correct components.
>
> **Q5: How does SAL-V perform when signal outputs are interdependent?**
>
> Thank you for the thoughtful question. As discussed in W1, we treat output signals as independent if their minimal and complete implementations do not directly rely on each other. Even if two signals share intermediate signal computations, they are considered independent as long as one does not consume the other’s value. This allows us to isolate and reward correct output signal code segments without being affected by incorrect ones.
>
> In cases where an incorrect signal depends on a correct one, SAL-V still performs well, since it isolates and rewards only the code segments related to correct signals. However, when a correct signal depends on an incorrect one, the reward becomes unreliable and such samples are not suitable for effective RL training. As shown in our W1 analysis, such cases are rare, only 4.8% of multi-signal samples, so we filter them out during training to ensure stability.
>
> **Q6: How reliable is AST extraction when the generated code is invalid or malformed?**
>
> Thank you for your valuable feedback. In fact, our AST extraction is applied only to syntactically valid Verilog code, all invalid or malformed code samples are filtered out during the construction of the signal-aware dataset. This ensures that the input to our signal extraction pipeline is always compilable, allowing reliable AST construction and rule-based analysis of signal-relevant code segments.
>
> To further evaluate the robustness of our method, we conducted a compilation-based validation experiment on all generated samples in RL training stage containing at least one correct signal. We extract the related code segment for correct signals in each sample and test their compilability using iverilog [1]. We observed 94.9% compile success rate among these extracted code, indicating that these extracted segments are both syntactically valid and compilable. These results confirm the reliability and scalability of our AST-based signal extraction methodology.
>
> [1] S. Williams, “Icarus verilog,” https://github.com/steveicarus/iverilog, 2000.

---

> > ### Comment · Reviewer_q7XC · 2025-08-04
> >
> > Thanks for the detailed and well-structured rebuttal. I appreciate how clearly you addressed each point.

---

> > > ### Author Response · Authors · 2025-08-04
> > >
> > > Thank you for taking the time to review our rebuttal. We sincerely appreciate your thoughtful and constructive feedback once again.

---

### Note · Authors · 2025-08-13

Dear Chair and Reviewers,

We sincerely thank the Area Chair and reviewers for their thorough evaluation and insightful comments that improved our manuscript.

We are delighted that the reviewers have recognized the signal-level reward in our work as "novel and well-justified, better, and a new paradigm" (q7XC, voDh, 2wdY), and that they found our AST analysis for identifying signal-aware code segments to be "novel, well-suited, and clever" (2wdY, EgJ3). Additionally, we are pleased that our method has been praised as "superior, robust, and broadly applicable" (voDh, 2wdY, EgJ3).

The main concerns raised by the reviewers focus on the comparison with the module-level reward method (q7XC, voDh) and whether our approach is effective when used with other RL algorithms beyond DPO (q7XC, 2wdY).

In our rebuttal, we compared our method with the module-level reward and demonstrated that the signal-level reward outperforms it. We also implemented SAL-V with GRPO and compared signal-level GRPO with module-level GRPO, showing that SAL-V is effective with on-policy RL methods and broadly applicable.

Reviewer q7XC raised concerns about the assumption of signal independence and the overhead of simulation and AST parsing. We explained the definition of signal independence in Verilog code generation, validated it experimentally, and reported that the overhead is negligible within the SAL-V framework.

Reviewer voDh raised concerns about a comparison with VeriPrefer. We compared the two in terms of reward granularity, correctness evaluation, and training paradigm, highlighting the advantages of our approach.

Reviewer EgJ3 expressed concern about possible data contamination in the CodeV dataset. We noted that decontamination was performed in the original paper and showed that our method outperforms CodeV under the same dataset and training setup, demonstrating its superiority.

Reviewers q7XC, 2wdY, and EgJ3 initially gave a score of 4, with two of them providing follow-up responses. Reviewer q7XC stated, “I appreciate how clearly you addressed each point,” and reviewer EgJ3 commented, “Thank you for addressing my concerns,” indicating that their concerns have been resolved. Reviewer voDh initially gave a score of 3 and noted, “My concerns have been addressed,” but has not yet updated final rating.

We sincerely thank the reviewers for their constructive feedback and the Area Chair for leading the discussions.

Best regards,

The Authors

---

### Decision · Program_Chairs · 2025-09-17

**Decision:**

Accept (poster)

**Comment:**

This paper presents SAL-V, a novel framework for Verilog code generation that shifts reinforcement learning optimization from module-level to signal-level rewards. By leveraging AST analysis and signal-aware verification, SAL-V extracts functionally correct code segments from partially incorrect modules, enabling more effective RL training via Direct Preference Optimization. Experiments show state-of-the-art performance on VerilogEval2.0 and RTLLM, with a 7B-parameter model matching much larger models and outperforming open-source baselines.

Reviewers highlighted the method’s originality, strong empirical results, and seamless integration with RL pipelines. Concerns included signal independence assumptions, runtime overhead, and generalizability. The authors’ rebuttal addressed these with empirical validation of signal independence, negligible runtime costs, and updated benchmark results.

SAL-V represents a significant step forward in automated HDL generation. Despite minor limitations, its novelty, technical rigor, and compelling results justify **acceptance**.